# Bridging Clinical Narratives and ACR Appropriateness Guidelines: A Multi-Agent RAG System for Medical Imaging Decisions

## Abstract

The selection of appropriate medical imaging procedures is a critical and complex clinical decision, guided by extensive evidence-based standards such as the ACR Appropriateness Criteria (ACR-AC). However, the underutilization of these guidelines, stemming from the difficulty of mapping unstructured patient narratives to structured criteria, contributes to suboptimal patient outcomes and increased healthcare costs. To bridge this gap, we introduce a multi-agent cognitive architecture that automates the translation of free-text clinical scenarios into specific, guideline-adherent imaging recommendations. Our system leverages a novel, domain-adapted dense retrieval model, ColBERT, fine-tuned on a synthetically generated dataset of 8,840 clinical scenario-recommendation pairs to achieve highly accurate information retrieval from the ACR-AC knowledge base. This retriever identifies candidate guidelines with a 93.9% top-10 recall, which are then processed by a sequence of LLM-based agents for selection and evidence-based synthesis. We evaluate our architecture using GPT-4.1 and MedGemma agents, demonstrating a state-of-the-art exact match accuracy of 81%, meaning that in 81% of test cases the predicted procedure set was identical to the guideline's reference set, and an F1-score of 0.879. This represents a 67-percentage-point absolute improvement in accuracy over a strong standalone GPT-4.1 baseline, underscoring the contribution that our architecture makes to a frontier model. These results were obtained on a challenging test set with substantial lexical divergence from the source guidelines. Our code is available at https://anonymous.4open.science/r/demo-iclr-B567/.

## 1 Introduction

Medical imaging procedure selection represents a critical decision point in patient care; clinicians must rapidly synthesize complex patient information with extensive clinical guidelines to determine the appropriate diagnostic procedure. The ACR Appropriateness Criteria (ACR-AC) is a set of guidelines designed to assist clinicians in choosing the most appropriate next imaging step or treatment for specific clinical scenarios (Subramaniam et al., 2019). This comprehensive compendium was first introduced in 1993, and now spans 257 diagnostic imaging and interventional radiology topics, covering >1,200 clinical variants and >3,700 clinical scenarios. The program mobilizes over 700 volunteer physicians across multidisciplinary expert panels and is revised annually. Each topic in ACr-AC is accompanied by a narrative synthesis, evidence tables, and a documented literature-search and rating process, with radiation-dose considerations and patient-friendly summaries to support use at the point of care. The sheer scale and ongoing curation of this knowledge base make its underuse in routine ordering decisions especially striking.

Despite their importance, these detailed guidelines are often underutilized in practice. Studies have found low adoption of ACR guidelines by physicians in multiple contexts. For instance, Bautista et al. (2009) found that only 1.59% (2 of 126 physicians) surveyed used the ACR appropriateness criteria as their primary resource when selecting imaging studies (Sheng et al., 2016; Bautista et al., 2009). Consistent with the limited uptake of formal criteria, utilization audits report substantial inappropriateness in day-to-day practice. In two Spanish public hospitals, only 47,8% of examinations involving ionizing radiation were deemed appropriate, whereas 31.4% were inappropriate.

The remaining fifth fell into categories such as repetition (4.6%), insufficient justification (8.4%), or absence of applicable guidelines (7.8%). Notably, in these hospitals, inappropriate procedures accounted for 19.6% of the total radiation dose delivered to patients and 25.2% of the associated costs, suggesting a substantial proportion of exposure and spending could be avoidable (Vilar-Palop et al., 2018). Adding to this, a multiregion Swedish analysis of 13,075 referrals assessed against ESR iGuide found that 37% of CT and 24% of MRI examinations were inappropriate, with no improvement in CT appropriateness over approximately 15 years; patterns varied by referrer, with CT ordered from primary care less appropriate than hospital requests (Ståhlbrandt et al., 2023). Furthermore, ordering of imaging procedures remains challenging; a significant number of imaging procedures may be unnecessary or yield no impact on patient outcomes (Francisco et al., 2021; Young et al., 2020), thereby increasing the risk of overdiagnosis which, in turn, leads to further avoidable side effects (e.g. potential contrast-related allergic reactions), increased burden on healthcare resources due to additional follow-up tests and treatments that do not enhance patient outcomes (Bautista et al., 2009) and the potential for negative effects on patients' mental health. This gap between available guidelines and actual clinical practice highlights the need for better decision support tools to bridge unstructured clinical narratives with structured guidance on imaging appropriateness criteria.

Recent advances in artificial intelligence offer a promising approach to address this challenge. Large Language Models (LLMs) such as GPT-4 have demonstrated remarkable capabilities in understanding complex medical texts and answering clinical questions at near-expert level (OpenAI, 2023). Additionally, medical-domain specific LLMs like MedGemma are trained on vast datasets of medical data, allows them to interpret and reason about medical data with enhanced nuance and accuracy, achieving or surpassing physician-level performance on medical benchmarks (Yang et al., 2025). However, LLMs have been shown to produce ungrounded statements, often referred to as "hallucinations", and, out-of-the-box, do not have access to up-to-date data. Retrieval Augmented Generation (RAG), powered by bespoke retrieval models, has emerged as an effective strategy to address such that issue (Lewis et al., 2020) thanks to their ability to efficiently retrieve information relevant to a query -which is then added to the context window of the LLM.

On their part, dense retrieval models have revolutionized information retrieval in the general domain. Models like ColBERT employ contextualized late interaction, encoding queries and documents into rich multi-vector representations that allow fine-grained token-level matching, this approach has been shown to substantially improve retrieval effectiveness for nuanced queries over traditional single-vector methods (Khattab & Zaharia, 2020). Applying dense retrieval to the medical domain is promising but non-trivial. Medical language uses many technical terms and abbreviations and is often ambiguous; relevant information (such as guideline text) may not directly match the free-form language of a patient's case presentation (Džuganová, 2019). Domain-specific adaptation of retrievers is therefore critical. Techniques such as fine-tuning with domain-specific supervision or unsupervised domain adaptation have been shown to yield significant gains in biomedical IR tasks. For example, Yüksel & Kamps (2024) demonstrate that adapting dense retrievers to a biomedical corpus improves retrieval effectiveness and makes the models attend more to domain-specific terminology.

However, even well-adapted retrievers and standalone LLMs face limitations in tackling complex, real-world clinical tasks: single-agent models may misinterpret ambiguous inputs, hallucinate unsupported conclusions, or fail to integrate multiple sources effectively (Ozmen & Mathur, 2025; Amugongo et al., 2025). In contrast, in multi-agent architectures each agent is tasked to complete focused jobs (such as retrieval, evidence validation, and response synthesis). Prior work has shown that structuring agent behaviour improves robustness, for instance, the ReAct framework demonstrated that interleaving explicit reasoning with targeted actions yields more interpretable and reliable performance (Yao et al., 2023). Building in this principle, recent work in multi-agent medical AI, such as MDAgents (Kim et al., 2024), automatically selects collaboration structures tailored to query complexity and demonstrates significant performance improvements over single-agent baselines in medical decision-making benchmarks, for instance, achieving best accuracy in 7 of 10 medical benchmarks and improving diagnostic accuracy by up to 9.5 percentage points on tasks like DDXPlus. Moreover, broader evaluations like MedAgentBoard further reveal that multi-agent collaboration enhances task completeness in clinical workflow automation, meaning that agents are more likely to generate usable outputs across multi-step processes (such as modeling code, visualizations, and reports) rather than failing partway through, although end-to-end correctness still remains a challenge (Zhu et al., 2025). Inspired by these works, we designed our system as a multi-

agent architecture to support the decision of a clinician who is given a patient scenario (e.g. "25 year old female showing clinically significant breast pain.") and is asked to determine the most appropriate procedure (e.g. Ultrasound of the breast).

In building our approach around dense retrieval we take inspiration from recent work utilizing LLMs and RAG for automated medical coding. For instance, Klang et al. (2024) demonstrated that applying RAG to emergency department notes increased coding accuracy in medical billing from 0.8% to 17.6%, also improving specificity compared to human coders. In Kwan (2024) the authors reported that on the task of predicting ICD-10-CM codes from single-term medical conditions, a Retrieve-Rank approach achieved 100% top one accuracy in a 100-case evaluation, far surpassing a plain GPT-3.5 baseline which reached only 6%. However, off-the-shelf models without domain-specific fine-tuning or retrieval augmentation showed substantial limitations, such as low agreement rates (approximately 15%) with human coders and frequent issues like over-extraction, where the model generates excessive or spurious ICD codes beyond what the clinical documentation supports, and imprecise outputs (Simmons et al., 2024; Ong et al., 2023). In contrast, several studies show that domain-specific fine-tuning or retrieval augmentation for medical coding significantly improved exact match accuracy, precision, and recall (Hou et al., 2025; Rollman et al., 2025). These findings underscore the critical importance of specialized fine-tuning and retrieval techniques for leveraging LLMs effectively in medical contexts.

Following in this line of research, in the following we document these contributions:

- **Domain-Adaptive Dense Retrieval:** We introduce and evaluate a domain-adaptive fine-tuning strategy for ColBERT, leveraging GPT-generated scenario-recommendation pairs. In 93.9% of our test cases, our ColBERT-based retriever correctly included the applicable ACR-AC guideline within the top-10 retrieved results. This is a substantial improvement over BM25 (35%) and GPT-4.1 without retrieval augmentation (48.6%). Our domain-adaptive fine-tuning strategy leverages GPT-generated scenario-recommendation pairs to achieve this high retrieval performance.

- **Effective Clinical Decision Support:** We demonstrate that our multi-agent system achieves 81% exact match accuracy (i.e. the predicted procedure set is identical to guideline's reference set; see §4.2 for details)(F1 = 0.879) on guideline-based imaging selection. This represents a 67-percentage-point absolute gain over a no-retrieval single-agent GPT 4.1 baseline (14% EM, F1 = 0.486) which underline the benefits of retrieval augmentation and multi-agent decision-making in clinical decision support.

## 2 DATA AND IMPLEMENTATION DETAILS

### 2.1 ACR GUIDELINES DATASET

We obtained from ACR the full collection of Appropriateness Criteria documents covering various clinical scenarios in diagnostic imaging. The ACR Appropriateness Criteria currently span 257 clinical topics, featuring more than 1,200 clinical variants and over 3,700 clinical scenarios. Each guideline typically defines a clinical condition (e.g. "Abdominal Aortic Aneurysm") and provides multiple variants, slight scenario qualifications or subgroups (such as different patient demographics or comorbidities). For each variant, the guideline lists a set of imaging procedures (X-ray, CT, MRI, etc.) with an appropriateness assessment: usually appropriate, may be appropriate and usually not appropriate. This hierarchical structure can be summarized as a 1:N:M mapping from conditions to variants to procedures (Figure 3). In our system, retrieval is trained to return the correct guideline variant, and the selection stage is evaluated on whether it recommends the procedures labelled "Usually Appropriate" for that variant. This allows the model to retrieve any of the clinically endorsed procedures, accommodating cases where multiple options are considered equally appropriate by ACR guidelines. The ACR-Appropriateness Criteria were parsed with the permission of the American College of Radiology (as granted on 22/01/2025), as reflected in the structured Hugging Face dataset by Menolascina (2024) (https://huggingface.co/datasets/fmenol/acr-appro-3opts-v2 to support reproducibility. We release our full synthetic dataset ("finetuning-colbert-acr-synthetic") of 8,840 scenario–variant pairs on Hugging Face (https://huggingface.co/datasets/

`satriopbd/finetuning-colbert-acr-synthetic`). This dataset derives from the ACR data under permission, and is intended for public use upon publication.

To better capture the variability of real-world documentation, we augmented the dataset by synthesizing eight clinical descriptions for each of the 1,105 scenario variants using the MedGemma-27B language model. These included three semantically graduated notes (progressing from close to more distant rewordings of the original phrasing) and five synonym-enriched notes (lexically diverse paraphrases). Importantly, this diversity was designed to mirror the way actual clinical documentation varies: physicians use different synonyms ("eye shaking" vs. "nystagmus"), abbreviations ("SOB" vs. "shortness of breath"), and shorthand styles, depending on specialty, training, and context. Figure 1 shows an example for the variant "Child with isolated nystagmus. Initial imaging."

Figure 1: Example of synthetic query generation. Example shown for "Child with isolated nystagmus. Initial imaging."

## 2.2 COLBERT FINE-TUNING

As initial tests revealed that general purpose retrieval/ranking models were poorly suited to the task of identifying relevant passages from existing guidelines, we were able to identify the likely cause of this in the heterogeneity/ambiguity of medical language. To address this issue we opted to fine tune ColBERTv2 (checkpoint (Santhanam et al., 2021)) on training pairs constructed from our dataset. Each training pair consists of a synthetic clinical description (query) and its corresponding ACR guideline variant (document). This framing teaches the model to map diverse, free-text patient notes to the canonical variant definitions in the guidelines. During training, the model learns to embed queries (e.g. "6 year old boy with rapid eye movement.") and guideline text ("Child with isolated nystagmus") such that the cosine similarity of relevant query–document pairs is higher than that of non-relevant pairs. We used RAGatouille's RAGTrainer with ColBERT, which employs a n-way classification approach where each query is paired with one positive document (the correct guideline) and several randomly sampled negatives (other guidelines from the corpus) for each query. Training was done for 10 epochs with a learning rate of 1e-5. The fine-tuning dramatically improved retrieval: on unseen test scenarios the correct ACR guideline was returned within the top 10 results for 93.93% of queries (recall@10) and already within the top 5 for 90.58% of queries (recall@5), giving the downstream LLM a very high likelihood of being returned relevant evidence.

## 2.3 ORCHESTRATION AND PROMPTING

The multi-agent pipeline is orchestrated sequentially in our implementation, the ACR retrieval agent takes the one-liner prompt from the user/clinician and processes it through our fine-tuned ColBERT against the indexed ACR knowledge base. ColBERT returns the top 10 most relevant results based on its learned representations of both the query and the candidate documents in the corpus. These 10 retrieved results are passed to a Large Language Model (LLM) selector agent, which identifies the single best-matching variant. Finally, the selected variant is mapped to its associated imaging procedures in the guideline, and the system outputs the set of procedures labeled "Usually Appropriate". The sequential orchestration of these components balances computational efficiency with

retrieval precision, ultimately delivering high-quality clinical information retrieval for healthcare practitioners.

## 3 METHODOLOGY

To address the challenge of mapping unstructured clinical notes to the most appropriate imaging procedures, we propose a multi-stage information retrieval and synthesis pipeline. Given a clinical scenario (e.g. "Female, 28 year old, presents with focal, noncyclical mastalgia", our methodology is designed to first identify the most relevant guideline from the ACR-AC (Condition: "Breast pain", Variant: "Female with clinically significant breast pain (focal and noncyclical). Age less than 30. Initial imaging.")  and then synthesize this information into a coherent, evidence-based response ("Ultrasound of the breast is *usually appropriate* when investigating clinically significant breast pain (focal and noncyclical) in a female patient under 30 year old). The architecture we proposed is composed of three sequential stages: (1) dense retrieval and ranking, (2) targeted selection, and (3) evidence-based synthesis, as illustrated in the overall system configuration diagram (Figure 5). In our system, each guideline variant is treated as a distinct document segment, which serves as the basic unit of retrieval and indexing. These segments correspond to clinically specific scenarios (e.g., "Child with isolated nystagmus. Initial imaging."), each linked to its associated set of procedures.

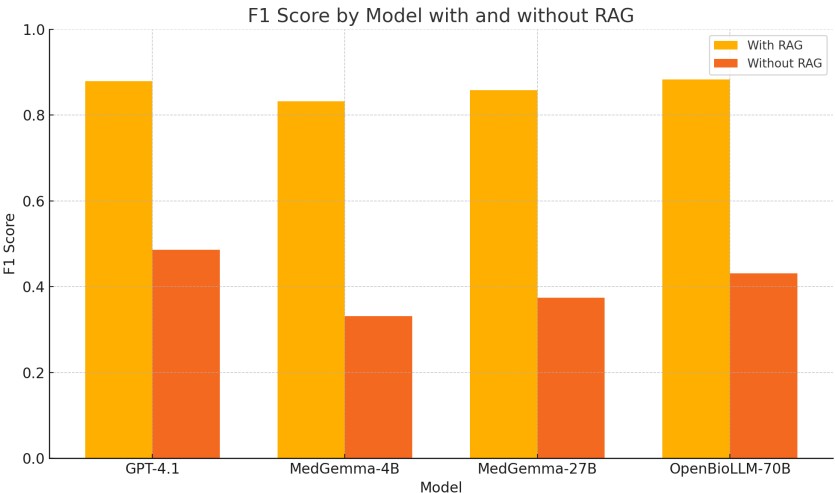

Figure 2: F1 Score Comparison Across Models With and Without RAG

In the first step, we use a ColBERT model fine-tuned on domain-specific clinical scenarios to retrieve relevant guidelines from the ACR Appropriateness Criteria and rank them based on their semantic overlap with the clinical scenario. Out-of-the-box ColBERT, while strong in general domains, struggles with the lexical ambiguity and specialized terminology of medical texts, leading to significantly lower recall. Fine-tuning with our synthetic pairs closes this gap and yields large improvements in retrieval effectiveness (see §3.4 for ablation results). This model processes user-supplied clinical scenarios (see Fig 5), to identify a set of candidate guidelines/recommendations. Subsequently, a separate "Selector" agent evaluates the retrieved candidates in the context of the original query (Fig. 1b). This agent is tasked with identifying the single best-matching guideline from the ranked list. In the final stage, a supervisor agent synthesizes the selected evidence into a guideline-aligned imaging recommendation, corresponding to the Output Module (D) in Figure 5, which takes the chosen variant and retrieves its associated procedures from the knowledge base.

Each of these components is discussed in detail in the following subsections: Section 2.1 describes the ColBERT-based retrieval mechanism, and and Section 2.2 covers the selector and supervisor agents.

## 3.1 COLBERT-BASED DENSE RETRIEVAL

Applying ColBERT directly to the medical domain is non-trivial, as clinical narratives are often complex, ambiguous, and full of specialized terminology and abbreviations. Out-of-the-box models tend to underperform because the same medical condition can be described in many different ways, and surface-level lexical overlap with guideline text is often low. To address this challenge, we fine-tuned ColBERT specifically for mapping clinical scenarios to ACR guideline text. The ACR Appropriateness Criteria (ACR-AC) are organized hierarchically: each condition (e.g., "Fibroids") consists of multiple variants representing clinically distinct scenarios (e.g., "Clinically suspected fibroids. Initial imaging." or "Known fibroids. Treatment planning. Initial imaging."). Each variant is linked to a set of candidate imaging procedures (e.g., ultrasound, MRI, CT), annotated with appropriateness ratings such as "Usually Appropriate", "May Be Appropriate", or "Usually Not Appropriate" (see Figure 3).

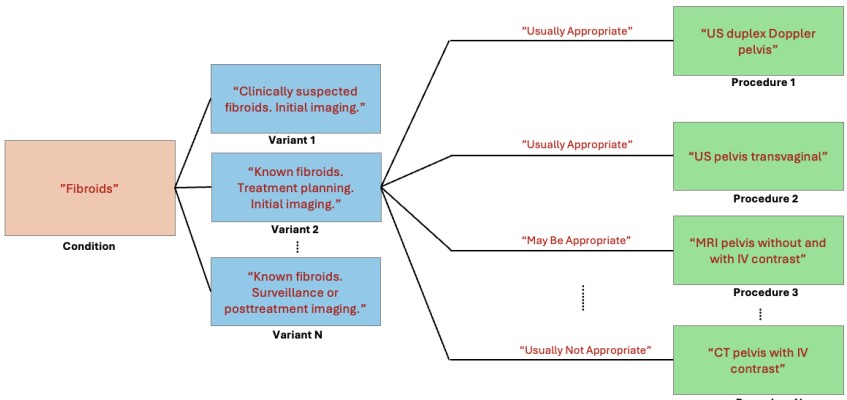

Figure 3: ACR-AC structure (with example)

Our fine-tuning employed a unique variant training approach to ensure comprehensive coverage of the ACR Appropriateness Criteria. We trained on a dataset comprising 1,105 unique guideline variants adapted from the Hugging Face dataset by Menolascina (2024), each expanded into eight synthetic clinical descriptions generated with MedGemma-27B (see Section 3.1 and Figure1), yielding 8,840 total training examples. The training configuration used a batch size of 8, a learning rate of 5e-6, and ran for 10,000 steps.

The query is encoded by the ColBERT model and searched against an index of ACR guideline segments (Figure 5, part B), which are represented using multiple token-level embeddings for late interaction matching.

The ColBERT late interaction mechanism enables fine-grained alignments between query terms and guideline content. For example, consider the query "6 year old boy with rapid eye movement". This is a non-technical description of the clinical phenomenon known as *nystagmus*. Even if the variant text in the ACR document uses terminology such as "nystagmus" or "vestibular disorder", ColBERT's late interaction mechanism enables the model to match semantically related terms like "dizziness" and "balance" across embeddings, improving recall over exact lexical matching.

## 3.2 LLM-BASED SELECTION

The ColBERT retrieval system described above returns the 10 most relevant guidelines from ACR-AC, along with metadata including guideline titles and their appropriateness ratings. A Large Language Model then serves as the final selector, processing the original query alongside the retrieved guidelines to produce a ranked recommendation with supporting rationale. We evaluated this approach using two LLM configurations: GPT-4.1 and MedGemma-4B, allowing us to assess how different reasoning capabilities affect the final selection quality while keeping the core retrieval pipeline consistent.

## 4 EXPERIMENTS AND RESULTS

We designed our evaluation to answer two main questions: (1) How well does the system's recommended imaging match the established ACR guideline recommendations? (2) What is the contribution of each component to the overall performance?

### 4.1 EVALUATION DATASET

As described in Subsection 2.1, we generated a test set of synthetic patient scenarios corresponding to various ACR guideline variants. Each scenario is a paragraph of a few sentences describing a patient's demographics, symptoms, relevant lab findings, and clinical context. The diversity of cases is high – e.g., one case might be "52-year-old male with soft tissue mass, suspecting recurrence." another "48-year-old male with musculoskeletal lump, possible cancer, monitoring for spreads into lungs." For each scenario, we have a ground-truth appropriate imaging procedure based on the ACR Appropriateness Criteria. In some cases, the guidelines list more than one appropriate option, we treated a recommendation as correct if it matched any of the top-tier options.

To ensure the evaluation dataset presents a realistic and challenging assessment, we deliberately constructed scenarios that require nuanced clinical reasoning rather than simple keyword matching. Compared to the true variant labels in the ACR guidelines, our synthetic patient scenarios exhibit a Jaccard similarity score of 0.088, indicating substantial lexical divergence that necessitates semantic understanding rather than surface-level text matching. Furthermore, a baseline BM25 retrieval approach achieves only a 35% top-10 hit rate on this dataset, demonstrating the difficulty of the task and validating our dataset's capacity to differentiate between sophisticated and naive retrieval approaches.

### 4.2 METRICS

We report **Exact Match Accuracy**, defined as the percentage of scenarios for which the set of procedures recommended by the system is identical to the set of 'usually appropriate' procedures in the ground-truth guideline. Partial overlap (e.g. recommending only one of several valid procedures) is not counted as an exact match. To capture such partial correctness, we additionally report the **F1-score**, which measures the degree of overlap between the predicted set and the ground-truth set when multiple procedures are valid. Additionally, to isolate the retriever's performance from the generator, we measure the **Retrieval Recall@K**, whether the correct guideline text was present among the top K documents given to the LLM. As noted, our ColBERT retriever achieved 93.9% recall at K=10, which means in only a few cases did the retrieval agent completely miss the relevant guideline.

### 4.3 ABLATION STUDY: EFFECT OF FINE-TUNING COLBERT

We compared the pre-trained (non-fine-tuned) ColBERT with ColBERT fine-tuned on increasing fractions of our synthetic training set (Fig. 4). On our test set, the pre-trained baseline achieved 69.9% Recall@10. Performance improved consistently with more in-domain data, reaching 92.1% Recall@10 at training fraction $f = 0.8$ and 93.9% at $f = 1.0$. Thus, while recall continues to improve, the incremental gains beyond roughly $f \approx 0.8$ are modest. A similar pattern holds for other cutoffs: Recall@1 rises from 39.3% (pre-trained) to 65.1% ($f = 1.0$), with most of the improvement accrued in $f \in [0.6, 0.8]$. These results indicate diminishing returns as additional examples increasingly reinforce patterns already captured by the model.

### 4.4 SYSTEM CONFIGURATION

We evaluated several configurations to our system, (A) using GPT-4.1 as the supervisor LLM, and (B) using MedGemma-4B as the supervisor. Both configurations used the same ACR retrieval components. We also compared several LLMs without RAG to establish baseline performance and assess the inherent capabilities of large language models on clinical imaging recommendations. Specifically, we tested the dataset on GPT-4.1, MedGemma-27B, and OpenBioLLM-70B to evaluate the capabilities of advanced LLMs, including those with specialization in the medical domain. This comparison allows us to understand whether retrieval augmentation provides meaningful improvements over the substantial parametric knowledge already present in state-of-the-art general-purpose

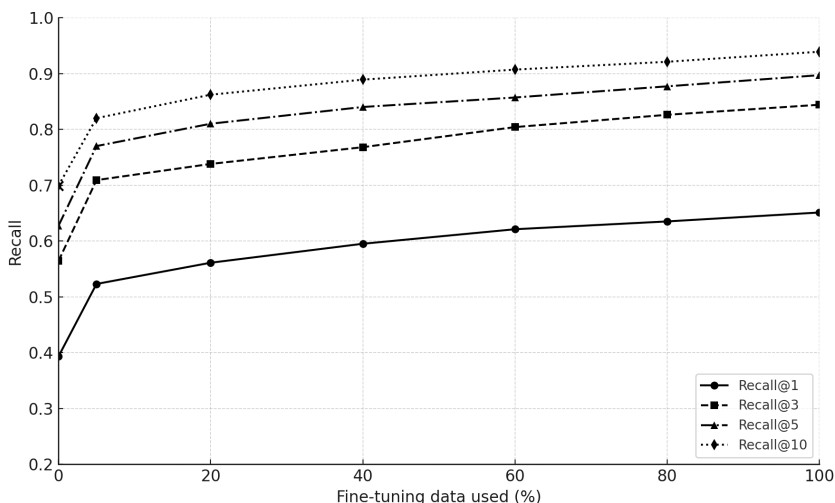

Figure 4: Ablation study on ColBERT fine-tuning

and medical-specialized language models, particularly when dealing with the nuanced clinical reasoning required for ACR guideline adherence.

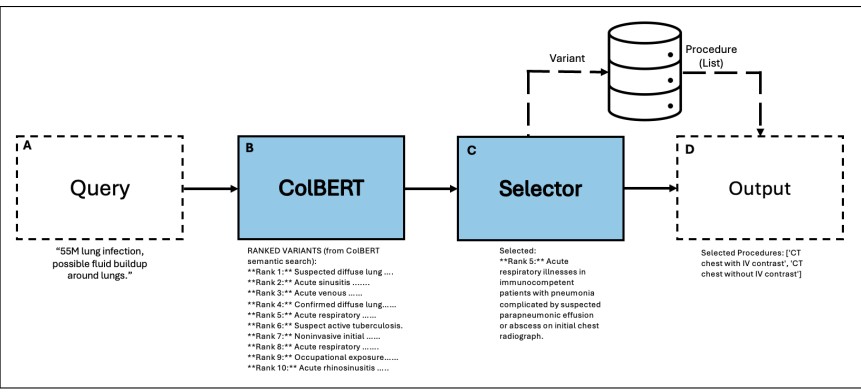

Figure 5: Overview of our multi-agent clinical decision support pipeline. A free-text clinical query (A) is encoded and semantically matched by the ColBERT retriever (B) against ACR guideline variants. The top-ranked variants are passed to an LLM-based selector (C), which chooses the most appropriate variant given the query context. The selected variant is then mapped to its associated set of imaging procedures (D), from which procedures labeled "Usually Appropriate" in the ACR Appropriateness Criteria are returned as the system output.

## 4.5 RESULTS

The GPT-4.1 powered system (A) achieved an Exact Match rate of 81% on the test scenarios, with a macro-averaged F1-score of 0.879. This means over 4 out of 5 test cases, an exact imaging procedure recommended by the ACR guidelines was identified as the top choice by the system. In the remaining cases, the recommendations were often partially correct (for example, the model might recommend an ultrasound where the guideline said ultrasound or MRI were equally appropriate, thus not counted as exact match but still appropriate). MedGemma-4B with RAG achieved 75% Exact Match (F1 = 0.832), while MedGemma-27B and OpenBioLLM-70B also performed well with RAG, reaching F1 scores of 0.858 and 0.883 respectively. However, without retrieval augmentation, performance dropped sharply across all models (e.g., GPT-4.1: F1 = 0.486; MedGemma-4B: F1 = 0.331). These trends, visualized in Figure 5, highlight the critical role of retrieval in grounding LLM recommendations with guideline-based evidence. Notably, smaller models with RAG often

outperformed larger models without it, emphasizing that access to relevant clinical context is more impactful than model size alone in this task.

Without RAG to retrieve relevant guidelines for context, all tested LLMs experienced substantial performance degradation across all metrics. This sharp decline underscores the critical importance of retrieval-augmented generation for grounding responses in verifiable clinical sources. In domains like healthcare, where mistakes can have serious consequences, anchoring generative outputs to trusted references is essential for maintaining safety, accountability, and clinician confidence. Notably, MedGemma-4B outperformed the much larger MedGemma-27B model when equipped with RAG, reinforcing the idea that access to relevant, domain-specific evidence can outweigh raw model size in complex decision-making. Beyond accuracy, retrieval augmentation also contributes to the trustworthiness of generative AI systems, which is a key factor for real-world adoption in clinical environments where transparency and evidence-based recommendations are critical.

## 5 DISCUSSION

Our findings demonstrate that a multi-agent RAG system, powered by a domain-finetuned ColBERT retriever, can effectively bridge the gap between unstructured clinical queries and structured ACR guidelines. The system's high accuracy (81% exact match; 0.879 F1 with GPT-4.1) confirms that this architecture can reliably translate brief, realistic clinician-style prompts into evidence-based imaging recommendations. This performance is particularly notable given the lexical diversity of our test set (0.088 Jaccard similarity), where simple keyword-based methods fail significantly (35% top-10 recall for BM25).

Our approach builds on recent work like accGPT (Rau et al., 2023) and Glass AI (Zaki et al., 2024), which also highlight the potential of RAG for this task. However, our primary contribution lies in the specialized, multi-step process that first maps a free-text query to a canonical ACR variant before retrieval and synthesis. This semantic bridging directly addresses the core challenge of guideline implementation: translating the messy reality of a patient's presentation into the formal ontology of the guidelines. By automating this translation with high fidelity, our system provides a direct mechanism to combat the documented underutilization of ACR criteria (Bautista et al., 2009) and reduce the risk of ordering inappropriate, low-yield imaging procedures (Francisco et al., 2021).

Despite these promising results, we acknowledge several limitations that frame our future work. First, our evaluation relies on synthetically generated patient scenarios. While designed to be challenging and realistic, they cannot capture the full complexity, ambiguity, and occasional errors present in real-world electronic health records. Second, our system is currently optimized for "one-liner" queries and may require adaptation to process longer, more detailed clinical narratives.

Future research will proceed along two main tracks to address these limitations.

1. **Clinical Integration and Validation:** The next critical step is to deploy the system in a "human-in-the-loop" framework with practicing radiologists. This will not only allow for evaluation on real clinical data but also help refine the model based on expert feedback, ensuring safety and building clinician trust. We also plan to expand the knowledge base to include a broader diversity of complex and atypical cases.

2. **Architectural Enhancements:** We will explore more dynamic multi-agent architectures where the system can adapt its retrieval and reasoning strategies based on query complexity. This could involve a meta-agent that allocates tasks or invokes specialized reasoning modules, further improving both efficiency and accuracy across a wider range of clinical contexts.

In conclusion, this work presents a robust and practical pathway for integrating advanced AI into routine medical imaging workflows. By focusing on the critical translation layer between clinical narrative and established guidelines, our system offers a tangible tool to enhance the quality, safety, and evidence base of clinical decision-making.

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
