# OpenReview forum: "Bridging Clinical Narratives and ACR Appropriateness Guidelines: A Multi-Agent RAG System for Medical Imaging Decisions"
_ICLR.cc/2026/Conference — Submitted to ICLR 2026_

### Official Review · Reviewer_zayp · 2025-10-26

**Soundness:** 3
**Presentation:** 2
**Contribution:** 2
**Rating:** 2
**Confidence:** 4

**Summary:**

This work studies a Text Retrieval Problem of retrieving ACR appropriateness criteria by clinical conditions (scenario). An 8,840 synthetic dataset is constructed, and an LLM-RAG system is proposed. Given a clinical condition as query, a set of variants are first retrieved by measuring the sematic similarity of the query and all variants with a fine-tuned ColBERT. An LLM selects a proper one from the returned variants as output, and an agent matches several recommended image exams.

A new scenario of text-to-text retrieval research and a new dataset in the medical domain. But novelty and contribution are limited. The practical challenge is described, but the research challenge is unclear to me.

**Strengths:**

This work studies a Text Retrieval Problem of retrieving ACR appropriateness criteria by clinical conditions (scenario). An 8,840 synthetic dataset is constructed, and an LLM-RAG system is proposed. Given a clinical condition as query, a set of variants are first retrieved by measuring the sematic similarity of the query and all variants with a fine-tuned ColBERT. An LLM selects a proper one from the returned variants as output, and an agent matches several recommended image exams.

Novelty: a new scenario of text-to-text retrieval research and a new dataset in the medical domain.

**Weaknesses:**

1. The novelty and contribution are limited, where no new model or algorithm is proposed, or new theories or theoretical proofs are proposed.
2. No research challenge is identifiied in this work, where the proposed problem seems easy to solve.
3. Limited comparison of other baselines or related works.
4. The introduction section contains too much statistical data, which makes the research questions unclear.
5. It is hard to say that the illustrated multi-agent system in this work really was ``multi-agent``. There are not many agents collaborated for solving the same problem. Instead, there is a workflow, where a LLM-based Selector selects one of candidate sentences retrieved by ColBERT.

**Questions:**

1. What is the ``selected evidence`` mentioned in the Line 265? How can the synthesized evidence retrieve the associated procedure of the chosen variant.

2. The work mentioned that the proposed architecture is composed of three sequential stages: (1) dense retrieval and ranking, (2) targeted selection, and (3) evidence-based synthesis. Which is the section or details of the ``evidence-based synthesis``?

3. Why is the performance of ``Without RAG`` so worse? If ``Without RAG`` represents the ColBERT-based retrieval module which learned by the ground-truth query-answer pair, then the lower scores seem to indicate that ColBERT is underfitted.

---

### Official Review · Reviewer_F261 · 2025-11-01

**Soundness:** 2
**Presentation:** 2
**Contribution:** 1
**Rating:** 2
**Confidence:** 4

**Summary:**

This paper presents a retrieval-augmented generation system for recommending medical imaging procedures based on the ACR Appropriateness Criteria. The authors fine-tune a ColBERT dense retrieval model on synthetically generated clinical scenarios, achieving 93.9% top-10 recall, and combine this with LLM-based selection agents to achieve 81% exact match accuracy on their test set. While the clinical problem is important and the technical execution is competent, the paper suffers from fundamental evaluation limitations that significantly undermine the validity and generalizability of the results.

**Strengths:**

1. Good explanation of ACR scale and motivation.
2. Fine-tuned ColBERT shows strong retrieval recall (93.9%).
3. Clear figures and consistent gain from retrieval vs. no retrieval.

**Weaknesses:**

1. The system is just a fixed RAG pipeline, not interactive or collaborative agents.
2. GPT-4.1 results seem unrealistically low; no fair prompt or comparison to simpler methods.
3. No tests on real clinical notes, no user study, no evidence that it works in practice.
4. We don’t know what kinds of mistakes happen or how serious they are.
5. No visualization of the final experiments result.

**Questions:**

1. Could you clarify why the system is described as multi-agent? From the paper, it appears to be a sequential RAG pipeline rather than multiple agents interacting or reasoning collaboratively.

2. The description of the experimental setup is somewhat confusing, especially how the synthetic test data were generated and how baselines were evaluated. Could you explain the data generation process and the prompting or evaluation details used for the baseline models?

---

### Official Review · Reviewer_JWbh · 2025-11-02

**Soundness:** 3
**Presentation:** 3
**Contribution:** 3
**Rating:** 6
**Confidence:** 3

**Summary:**

The authors propose and implement a multi-agent architecture for the assessment of guideline adherence. They generate large numbers of synthetic cases from ACR guidelines, and then assess the ability of their multi-agent system to determine the appropriate guideline.

**Strengths:**

- I commend the authors for choosing a relatively novel and important area for exploration and implementation with their system. Appropriateness gaps are an excellent prospective use case for artificial intelligence.
	- I appreciate the use of both a closed source frontier model, and a novel opensource model.
	- The introduction is very clear, and strongly establishes the importance and positioning of the authors' work.
	- Combination of both LLM and BERT-based methods is novel and interesting. The authors' description and use of the ColBERT system is clever and appears to be effective.

**Weaknesses:**

• The authors' synthetic data generation is useful, however I am not fully confident in relying on such systems without human validation. It is not clear to me if this was performed, although the paper does imply that this is a future direction. I think the lack of this is the most significant limitation of the paper. Appropriateness is a complex, clinically specific assessment that often relies on nuanced clinical factors. It is very possible that there may be changed made by the language model in the synthetically generated cases which although seen as benign by the model end up very substantially changing the clinical nature of the case. Without high-quality physician validation, this remains very limited as a benchmark (see e.g. https://ai.nejm.org/doi/full/10.1056/AIe2500143)
	• Similarly, the use of LLM-generated synthetic data to then be assessed by another LLM is a bit of a self referential loop. I worry that, at times, there may be subtle clues in the formation of the synthetic data which tip off the initial appropriate guideline in ways that are not "fair" to how these guidelines map onto the nuance of clinical practice. If an LLM turns guideline statement X (e.g. child with nystagmus) into some formulation Y (6M presents with shaky eyes), then that exact mapping is, by definition, already known to the LLM. Inverting this alone does not necessarily tell us whether the model would be able to effectively map the real patients in all of their messy complexity (where the real imaging-related complaint is actually a secondary issue in the chart, for example). In order to truly be confident here, I believe the paper would require either (ideally) the use of real-world data as a seed, or the use of human-generated synthetic data as a source of a bit more complexity.
Similarly, guideline recognition is an important first step, but much more needs to be done in order to assess the appropriateness of how these guidelines are implemented by the models in practice. The devil is very much in the details in these contexts. The current work is interesting, but very limited. I do, however, feel the authors are quite clear in acknowledging these limits.

**Questions:**

What, if any, validation was performed of the synthetic data generation? Were any explicit cross-model robustness checks (e.g. generating on different models than testing)?

Was any physician auditing of the text corpus performed?

---

### Meta-Review · Area_Chair_2Moh · 2026-01-10

**Summary:**

All three reviewers agree the clinical problem is important. My decision is driven by concerns about validity and generalizability of the evaluation. The core results are obtained on a fully synthetic dataset generated from the same guideline source the system is meant to retrieve from, with no clinician validation of the synthetic cases and no evaluation on real clinical notes or realistic workflows. Reviewers also question the framing and contribution: the method reads as a sequential RAG pipeline rather than a multi agent or interactive system, and the novelty is primarily in applying standard components to a new domain with a new synthetic dataset. Several reviewers also flag missing or unclear experimental details (how synthetic train and test are produced, baseline prompting, what error types occur, and what the evidence based synthesis stage contributes), which makes it hard to interpret the claimed gains and whether the GPT 4.1 baseline is a fair comparator.

Given these issues, I do not think the current submission supports the strength of the claims about guideline adherence or practical imaging decision support.

**Reviewer Concerns:**

No rebuttal was submitted.

**Reviewer Scores:**

No rebuttal was submitted.

---

### Decision · Program_Chairs · 2026-01-26

Reject